# Tumour Genome Characterization of a Rare Case of Pulmonary Enteric Adenocarcinoma and Prior Colon Adenocarcinoma

**DOI:** 10.3390/jpm11080768

**Published:** 2021-08-04

**Authors:** Robert J. Smyth, Valentina Thomas, Joanna Fay, Ronan Ryan, Siobhan Nicholson, Ross K. Morgan, Liam Grogan, Oscar Breathnach, Patrick G. Morris, Sinead Toomey, Bryan T. Hennessy, Simon J. Furney

**Affiliations:** 1Department of Molecular Medicine, Royal College of Surgeons in Ireland, D02 YN77 Dublin, Ireland; robertsmyth23@gmail.com (R.J.S.); joannafay@rcsi.ie (J.F.); liamgrogan@beaumont.ie (L.G.); osbreathnach@beaumont.ie (O.B.); patrickmorris@beaumont.ie (P.G.M.); sineadtoomey@rcsi.ie (S.T.); 2Department of Medical Oncology, Beaumont Hospital, D09 V2N0 Dublin, Ireland; 3Genomic Oncology Research Group, Department of Physiology & Medical Physics, Royal College of Surgeons in Ireland, D02 YN77 Dublin, Ireland; valentinathomas@rcsi.ie; 4Centre for Systems Medicine, Royal College of Surgeons in Ireland, D02 YN77 Dublin, Ireland; 5Department of Histopathology, St James’s Hospital, D08 NHY1 Dublin, Ireland; RRyan@STJAMES.IE (R.R.); SNICHOLSON@STJAMES.IE (S.N.); 6Department of Respiratory Medicine, Beaumont Hospital, Dublin and Royal College of Surgeons of Ireland, D02 YN77 Dublin, Ireland; rossmorgan@beaumont.ie

**Keywords:** pulmonary enteric adenocarcinoma, colorectal adenocarcinoma, whole-exome sequencing, intratumour heterogeneity, mutational burden

## Abstract

Pulmonary enteric adenocarcinoma (PEAC) is a rare variant of lung adenocarcinoma first described in the early 1990s in a lung tumour with overlapping lung and small intestine features. It is a rare tumour with fewer than 300 cases described in the published literature and was only formally classified in 2011. Given these characteristics the diagnosis is challenging, but even more so in a patient with prior gastrointestinal malignancy. A 68-year-old Caucasian female presented with a cough and was found to have a right upper lobe mass. Her history was significant for a pT3N1 colon adenocarcinoma. The resected lung tumour showed invasive lung adenocarcinoma but also features of colorectal origin. Immuno-stains were strongly and diffusely positive for lung and enteric markers. Multi-region, whole-exome sequencing of the mass and archival tissue from the prior colorectal cancer showed distinct genomic signatures with higher mutational burden in the PEAC and very minimal overlap in mutations between the two tumours. This case highlights the challenge of diagnosing rare lung tumours, but more specifically PEAC in a patient with prior gastro-intestinal cancer. Our use of multi-region, next-generation sequencing revealed distinct genomic signatures between the two tumours further supporting our diagnosis, and evidence of PEAC intra-tumour heterogeneity.

## 1. Introduction

PEAC is a rare variant of invasive adenocarcinoma first described in the 1990s [1]. It is defined as a pulmonary adenocarcinoma with an enteric component of at least 50% [2]. There are fewer than 300 cases described in the published literature with a prevalence of approximately 0.5% of all non-small cell lung cancers [3]. Given its rarity, distinct immunohistochemical and molecular features have yet to be identified, and a clear cut-off based on the percentage of positive neoplastic cells and/or the intensity of the immunostaining has yet to be established. Therefore, the possibility of co-occurrence of a gastrointestinal tumour must be strongly considered and investigated through standard diagnostic methods. At present, positive staining for at least one colon marker is necessary to confirm the diagnosis, with lung adenocarcinoma features such as TTF-1 and enteric markers CK20 and CDX2 often co-expressed. A review of one of the largest case series of 28 cases found that patients with PEAC compared to invasive lung adenocarcinoma tended to be older and male, with larger lesions and at a more advanced stage of disease [3]. They also determined that CK7 was a valuable marker in those who were TTF negative, as TTF-1 expression is frequently lost in non-small cell lung cancer with mucinous or enteric differentiation. A study of 129 PEAC cases compared to 50 colon cancers determined an intermediate sensitivity (71%) and specificity (82%) for the combination of CK7 and CDX2 positivity for PEAC identification [4]. Nottegar et al. found uniform co-expression of CK7 and CDX2 in all analysed samples (*n* = 46), even in samples from diagnostic biopsies [5]. Lung cancer research over the past number of decades has directly led to dramatic improvements in survival rates [6]. The main driver of this success has been the ability to characterize lung cancers based on their molecular signature and allow patients to avail of targeted therapies [7]. Since 2013 it has been the standard of care to test for tumour characteristics occurring at the cellular level in patients with advanced lung cancer, using next-generation sequencing or mutation-specific polymerase chain reaction (PCR) testing [8]. However, these benefits have now shown promise in patients with earlier stage disease, e.g., the ADAURA study [9], which may lead to a future prospect of all lung cancers being molecularly profiled representing a fully personalized genomic approach for all patients. However, few genomic studies have been performed to further characterize the molecular characteristics of PEAC tumours. Given the similarities in histology and immunohistochemical markers between PEAC and colorectal cancer we report the use of multi-region, whole-exome sequencing of a patient’s resected pulmonary enteric adenocarcinoma and archival tissue from her prior colon adenocarcinoma to help clarify the diagnosis and investigate somatic mutations and mutational heterogeneity.

## 2. Materials and Methods

### 2.1. Sample Processing and Sequencing

DNA was extracted from FFPE samples from two geographically distinct areas from the patient’s CR (CR_1-2) and from four quadrants of the patient’s lung tumour (LU_1-4) and the involved node (LU_ND), and a blood sample was taken for germline analysis. For all samples, exome capture was performed on 2 μg DNA using an Agilent SureSelect Human All Exome V3 kit (Agilent Technologies, Santa Clara, CA, USA). Samples were sequenced to a mean coverage of 83× using 91bp paired end reads on the Illumina HiSeq 2000.

### 2.2. Whole-Exome Sequencing Data Analysis

The quality of the FASTQ files generated was determined using FastQC, and adapter and primer sequences and low quality 3′ end reads were trimmed off using Trimmomatic [10]. Remaining reads (with a minimum length of 75 bases) were aligned to the hg19 reference genome using BWA [11]. PCR duplicates were marked using Picard Tools (http://broadinstitute.github.io/picard version: 2.9.2, accessed on 2 June 2021), and InDel realignment and base quality recalibration were conducted with GATK v3 [12]. Somatic single nucleotide variants (SNVs) were identified with mutation calling algorithms MuTect v1 [13] and somatic indels with VarScan 2 [14]. Identified somatic variants were annotated using Variant Effect Predictor [15] and variants within the targeted capture genes were kept for further analysis.

### 2.3. Copy Number Analysis

Copy number alterations and tumour purity and ploidy were estimated using Sequenza v3 [16], and annotated with ANNOVAR v2019Oct24 [17].

### 2.4. Mutational Signature Analysis

Mutational signature analysis was performed to inform on the exposures and biological history of a cancer. Mutational signatures were identified from SNVs using the R package deconstructSigs v1.9 [18] based on the pan-cancer catalogue of single base substitution (SBS) signatures referenced in the COSMIC v3 database (https://cancer.sanger.ac.uk/signatures/version version: 3, accessed on 2 June 2021).

## 3. Case Presentation and Results

A 68-year Caucasian female, former smoker with a 15-pack year history, presented with a new persistent cough. She had a past medical history of a pT3N1 colon adenocarcinoma treated with complete resection and adjuvant chemotherapy 13 years previously. She remained well with no evidence of tumour recurrence on interval colonoscopies or surveillance imaging. A CT chest and PET-CT were performed showing a 4.6 cm × 2.9 cm mass in the right upper lobe with an SUV_max_ of 9.2 (Figure 1A), without evidence of mediastinal nodal metastasis or extra-thoracic uptake. CT-guided transthoracic fine-needle aspirate was performed (Figure 1B), showing a moderate cellularity sample suggestive of primary lung adenocarcinoma with a TTF-1 positive stain (Figure 1C,D).

She underwent resection of the tumour and was noted to have direct nodal extension of the tumour into the right paratracheal node. The tumour histology showed invasive lung adenocarcinoma, but also large acini with central necrosis, extensive cribriform, and solid growth patterns that resembled colorectal carcinoma (Figure 2A–D). Immuno-stains were strongly and diffusely positive for CK7 and TTF-1 and some CK20 and CDX2 (Figure 3A–D). We compared this tumour to the patient’s prior colorectal carcinoma, which was an invasive, moderately differentiated adenocarcinoma with focal mucinous differentiation. It was positive for enteric markers CK20 and CDX2 (Figure 4A,B) and absent of lung markers CK7 and TTF-1 (Figure 4C,D).

A diagnosis of pulmonary enteric adenocarcinoma was made. Given the rarity of this tumour and her prior history of colon cancer, the possibility that this tumour represented an unusual presentation of a colorectal metastasis to the lung was considered with obvious ramifications to the patient for treatment and prognosis. To further investigate the characteristics of the tumour, we compared the mutational landscape of the lung tumour (LU) to that of the prior colorectal adenocarcinoma (CR).

DNA was extracted from FFPE samples from two geographically distinct areas from the patient’s CR (CR_1-2) and from four quadrants of the patient’s lung tumour (LU_1-4) and the involved node (LU_ND), and a blood sample was taken for germline analysis. For all samples, exome capture was performed on 2 µg DNA using an Agilent SureSelect Human All Exome V3 kit (Agilent Technologies, Santa Clara, CA, USA). Samples were sequenced to a mean coverage of 83× using 91bp paired end reads on the Illumina HiSeq 2000. Somatic single nucleotide variants (SNVs), short insertions and deletions (indels) and copy number alterations (CNAs) were identified by comparing each tumour sample and involved lymph node with the blood sample as a matched normal, as previously described [19].

We identified a higher mutational load in the CR samples, with an average of 1813 SNVs (1680–1945) and 616 indels (320 to 912), compared to an average of 495 SNVs (427–552) and 32 indels (28–40) in the LU samples (Figure 5A). Mutational burdens in the LU_ND sample were also significantly lower (410 SNVs, 25 indels). Less than half (37%) of the SNVs found in the multi-region sequencing of the LU tumour were common to all samples and only 12% of indels. Heterogeneity was evident within the two CR lesions, with an overlap of 59.3% SNVs and 5.1% indels, respectively. Between the two tumour types, 9.3% of SNVs were found to be common (Appendix A). Therefore, the tumour mutational burden was much higher in the CR tumour (range: 9.8–21.3 mutations per Mb) compared to the LU tumour (range: 5.3–6.4 mutations per Mb). There was also a definite distinction in candidate somatic driver mutations between the CR and the LU tumour (Figure 5B). The CR samples shared several driver mutations, including a common *BRAF* V600E mutation. Most candidate driver mutations found in the LU were shared by all tumour regions, notably a *KRAS* G12C mutation, which was also detected at a lower variant allele frequency (<3%) in the involved node (Figure 5B).

Distinct profiles emerged from the mutational signature analysis distinguishing the CR from the LU. The characterizing feature of the LU samples was the presence of a tobacco smoking-related signature (SBS4) in all four tumour samples and to a lesser extent in the lymph node sample (Figure 5C). A mismatch repair deficiency signature and an increased indel burden were evident in CR_1 and CR_2 (Figure 5A,C). We identified potential driver copy number alterations in the tumours (Figure 5D). While *KRAS* was amplified in all tumours, significant heterogeneity was observed both between and within the CR and LU tumours. CR_2 displays *DCC*, *FBXW7*, *SMAD4*, and *TP53* losses that were not evident in CR_1. All LU tumours and LU_ND showed a loss of *CDKN2A* and gain of *MYC.* However, we observed potential heterogeneity in the copy number states of *TP53*, *FBXW7*, and *CCNE1* in the LU samples.

## 4. Discussion

Few studies have investigated the molecular profile of PEAC, primarily from single-gene or a panel of few genes and usually focusing only on the major driver mutations of NSCLC. Lin et al. performed targeted sequencing on seven PEACs with frequent mutations in *ALK/ROS-1* (71%) and *TP53* (57%) [20]. Conflicting data regarding the relevance of the *KRAS* mutation has been seen. Wang et al. showed its presence in only 1 of 9 PEAC samples, whereas a later study by Nottegar et al. identified *KRAS* in 4/8 samples [21,22]. The largest study to date by Chen et al. analysed 129 cases of PEAC, with *KRAS* identified as the most common mutation (48%), whereas no mutations were detected in *EGFR* or *BRAF* [4]. We observed both *KRAS* mutation and amplification in the PEAC in this study. Mutational burden analysis using ultra-deep targeted sequencing of 13 PEACs and 5 CRCs showed a tendency towards a lower mutational burden in PEACs [23], although other published analysis has shown the opposite [4]. Despite evidence of smoking-related DNA damage in the PEAC in our study, all regions exhibited a lower mutational burden than the CR samples, which displayed genomic features of microsatellite instability [20].

Interestingly ethnic considerations may be important to consider when assessing the characteristics of these tumours. Palmirotta et al. in their review of the published literature of 259 analysed patients found most patients with PEAC to be Asian (62.7%). IHC expression of TTF-1, Napsin A, and SP-A were similar between patients diagnosed in Asian vs. European/North American countries, but significant differences were seen in the proportion of patients expressing CK7, CDX2, and CK20, with higher rates of CK20-positive tumours in the Asian population and CK7 and CDX2 being more significantly expressed in PEAC samples from Europe/North America [24]. Furthermore, statistically significant genomic differences were also seen, with *KRAS* mutations being more common in Europe/North America (60.2% vs. 10.8%) and *EGFR* mutations more common in Asian patients (23.0% vs. 1.2%) [24]. Furthermore, statistically significant genomic differences were also seen with *KRAS* mutations being more common in Europe/North America (60.2% vs. 10.8%) and *EGFR* mutations more common in Asian patients (23.0% vs. 1.2%). CDX2 is a homeobox gene encoding transcriptional factors for intestinal differentiation and in adults is expressed in the nuclei of intestinal epithelial cells. As mentioned previously, it is used in pathology as a tissue biomarker for intestinal morphology in patients with CRC and in PEAC. A recent meta-analysis of over 6000 patients with CRC showed that the level of CDX2 expression in stage II and III disease proved to be a strong prognostic factor, leading to 70% lower risk of death [25]. However, to our knowledge no such data exists for patients with PEAC.

Despite the increasing recognition of this tumour, understandable clinical concerns associated with the potential for this to represent a colorectal metastasis remain. This case demonstrates the diagnostic dilemma of managing patients with rare lung tumours. The patient’s history of colon cancer added an extra level of complexity to the diagnosis. This problem is nicely detailed by Jurmesiter et al., who showed that despite a blinded review by five senior pathologists of 15 PEAC and four metastatic colorectal carcinoma samples, in all cases at least one pathologist arrived at the wrong diagnosis with error rates ranging from 20–60% [26]. Our analysis showed important differences in the genomic landscape of these two tumours, particularly in terms of mutational burden, lack of shared SNVs, distinct mutational signatures, and copy number alteration heterogeneity.

## 5. Conclusions

Personalized medicine has become indispensable in modern medicine and this is perhaps most evident in the field of medical oncology. Lung cancer is arguably one of the most successful examples of this approach, where the discovery of molecular alterations in patients’ tumours has allowed for the development of targeted therapies. This has led to improved survival, better tolerability, and the ability to prognosticate based on the specific molecular pattern of the tumour. Recently, a targeted therapy was approved for patients with lung cancer harbouring the *KRAS* G12C mutation, the first successful treatment for *KRAS* mutated lung cancers. This personalized approach is advancing even further with the ability now to genotype a patient’s lung cancer using cell free DNA in patients’ plasma or “liquid biopsies” [27], allowing multiple samples to be taken longitudinally on treatment to detect dynamic changes over time.

To our knowledge, this is the first report of the use of whole-exome sequencing in the examination of PEAC. Significantly, the use of exome sequencing contributed an additional personalized dimension to this case and has facilitated a more accurate and reliable diagnosis. Although the *KRAS* G12C mutation was uniformly present in the tumour, less than half of SNVs were found in all four geographically distinct samples. The *KRAS* mutation was detected in the involved paratracheal node at a low variant allele frequency. Therefore, genomic spatial heterogeneity must be considered in the molecular evaluation of these rare tumours. Given the small numbers of cases published, interpreting the data is difficult. Much could be gained from a central registry to improve our understanding of this rare cancer.

## Figures and Tables

**Figure 1 jpm-11-00768-f001:**
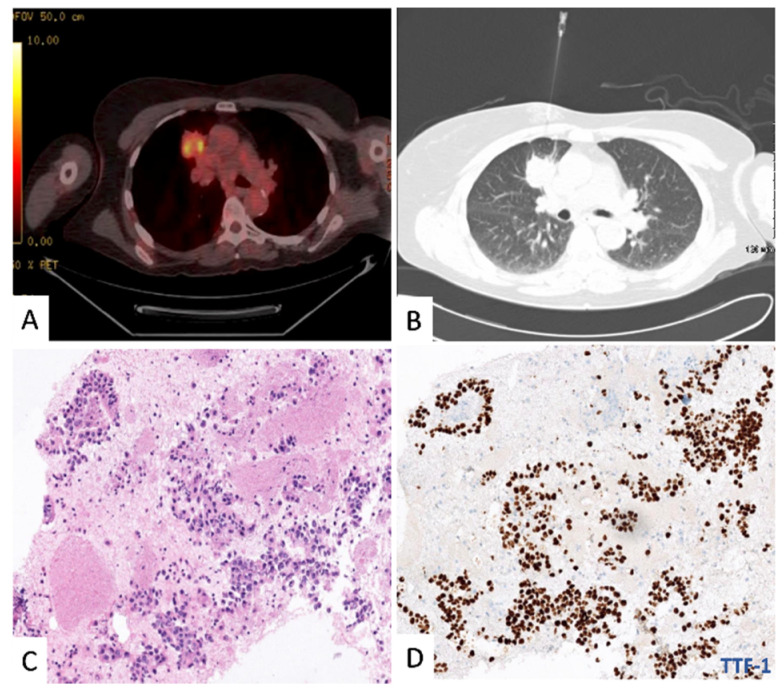
(**A**) PET-CT image of a right upper lobe mass (4.6 cm × 2.9 cm) with SUVmax of 9.2. (**B**) Initial diagnostic biopsy–20 GCT guided transthoracic needle aspiration of the lung mass. (**C**) Diagnostic biopsy H&E stain. (**D**) Diagnostic biopsy TTF-1 stain.

**Figure 2 jpm-11-00768-f002:**
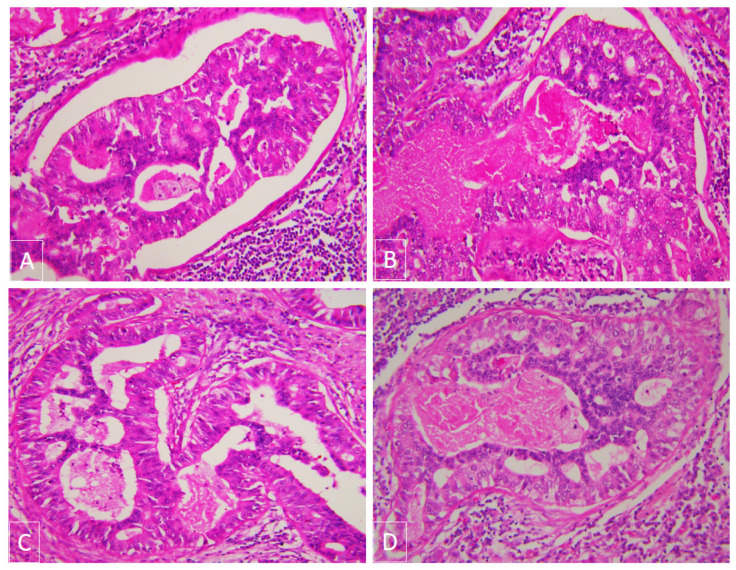
H&E: (**A**) and (**B**) adenocarcinoma, enteric pattern with large complex glands, (**C**) cribiform and solid growth patterns, plus (**D**) central necrosis.

**Figure 3 jpm-11-00768-f003:**
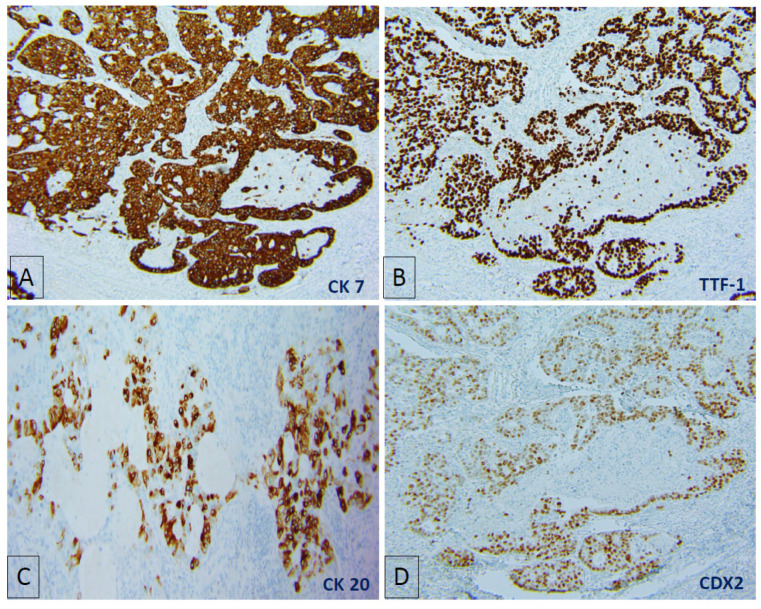
Immuno-stains showing strong and diffusely positive CK7 (**A**) and TTF-1 (**B**), with some CK20 (**C**) and CDX2 (**D**) expression.

**Figure 4 jpm-11-00768-f004:**
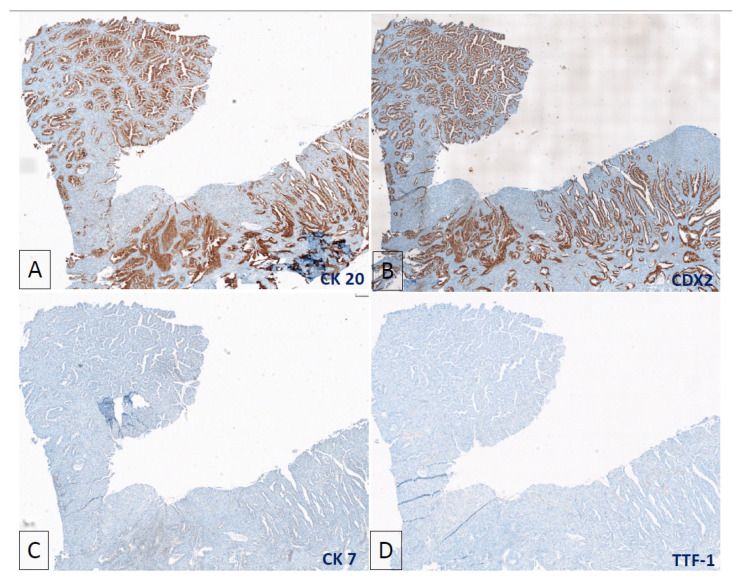
Immuno-stains of the previous colorectal carcinoma showing (**A**) strong and diffusely positive CK20 and (**B**) CDX2, and absence of CK7 (**C**) and TTF-1 (**D**).

**Figure 5 jpm-11-00768-f005:**
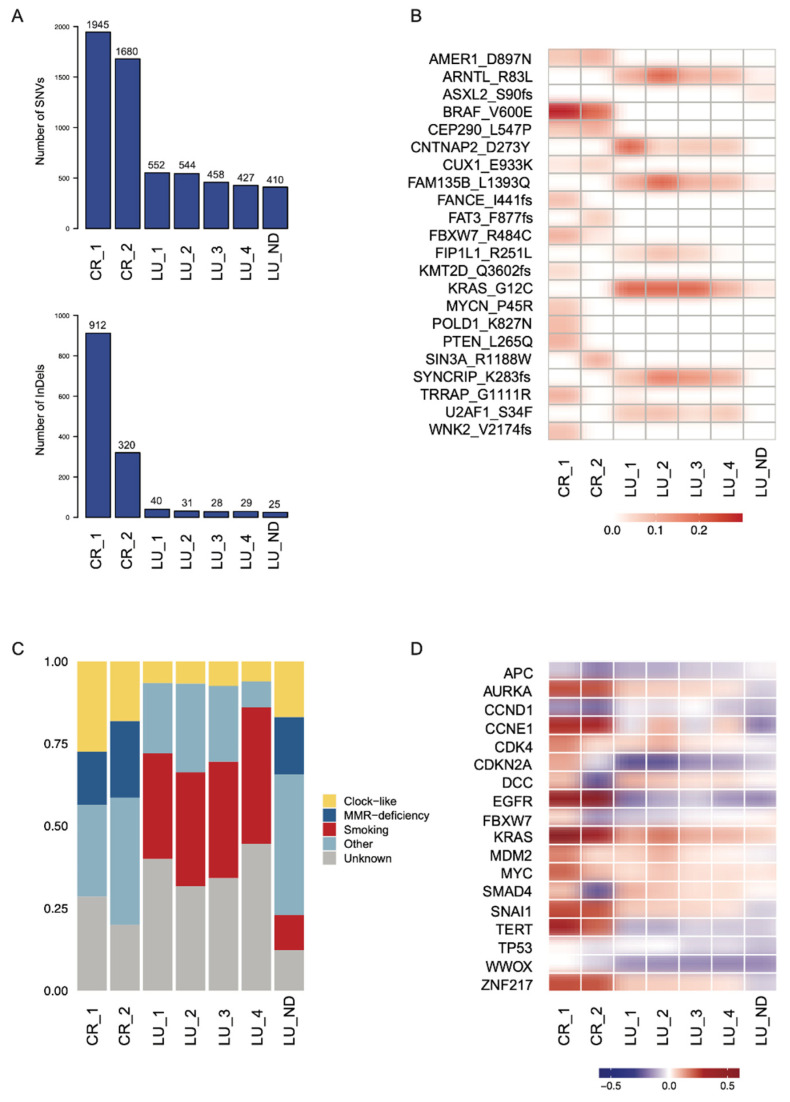
(**A**) Mutational burdens of SNVs and indels, quantified by whole-exome sequencing. (**B**) Variant allele frequencies of known and likely oncogenic mutations. (**C**) Tumour mutational profiles of total SNVs, defined by the weighted contributions of each input reference signature from COSMIC and identified by whole-exome sequencing. Signatures found in the samples include “clock-like” signatures (SBS1 and SBS5), “MMR-deficiency”-related signatures (SBS15, SBS21, and SBS26), the “smoking” signature (SBS4), and “other” signatures (SBS10b, SBS12, SBS16, SBS24, SBS27, SBS37, SBS49, and SBS50). (**D**) Median log-ratios of putative driver CNAs. Positive log-ratios (gains) are represented in red and negative log-ratios (losses) are represented in blue.

## Data Availability

The data presented in this study are available on request from the corresponding author. The data are not publicly available due to privacy restrictions.

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
