# Peer review of "Tumour Genome Characterization of a Rare Case of Pulmonary Enteric Adenocarcinoma and Prior Colon Adenocarcinoma"

_jpm, 2021, doi:10.3390/jpm11080768_

Round 1
Reviewer 1 Report
The authors have presented an interesting case that can help in the histopathological diagnosis. The case is well structured. However, there are several talking points:
-The authors should give a more focused character to personalized medicine and the possible implementation for a more precise screening and diagnosis.
-The authors must include better quality histological images. Authors should focus the samples.
-The authors must show an image where the nucleus can be seen at higher magnifications. It is important that the structure is shown.
-The authors should show the activity of Rb1, pRB and KI67. These markers are essential to give cohesion.
-What is the opinion of the authors regarding COX2? These points must be discussed by the authors in the manuscript in relation to survival for the results obtained to be applicable.
-In figure 2A the authors must show the deviations. Show the values.
-The conclusions must be focused towards a precision diagnosis.
Author Response
Reviewer 1
The authors have presented an interesting case that can help in the histopathological diagnosis. The case is well structured. However, there are several talking points:
- The authors should give a more focused character to personalized medicine and the possible implementation for a more precise screening and diagnosis.
We have added relevant information to the Introduction to show how lung cancer diagnosis has become more personalized due to molecular targeted therapy.
“Lung cancer research over the past number of decades has directly led to dramatic improvements in survival rates [6]. The main driver of this success has been the ability to characterize lung cancers based on their molecular signature and allow patients to avail of targeted therapies [7]. Since 2013 it has been the standard of care to test for these tumor characteristics occuring at the cellular level in patients with advanced lung cancer, using next generation sequencing or mutation specific polymerase chain reaction (PCR) testing [8]. However, these benefits have now shown promise in patients with earlier stage disease e.g. the ADAURA study [9] which may lead to a future prospect of all lung cancers being molecularly profiled representing a fully personalized genomic approach for all patients.”
In addition, we have expanded the Discussion and Conclusions to further personalize the manuscript. To emphasise this, we have added a new Figure 1 which shows imaging and pathology slides of the initial diagnostic biopsy. In addition, we have added a new Figure 4 of the historic colon adenocarcinoma immunostaining.
- The authors must include better quality histological images. Authors should focus the samples.
Our pathology colleague and co-author, Dr. Nicholson has prepared new high power, sharper histological images, which we include in a new Figure 2.
- The authors must show an image where the nucleus can be seen at higher magnifications. It is important that the structure is shown.
As mentioned above, Dr. Nicholson has prepared new high power, sharper histological images to the limits of the microscope camera, which we include in a new Figure 2. We hope that these new images are sufficient to address the reviewer’s concern.
- The authors should show the activity of Rb1, pRB and KI67. These markers are essential to give cohesion.
We respect the reviewer’s comment. However, Ki-67 is not used routinely in lung adenocarcinoma as it has high expression in this cancer subtype, and its utility in prognosis and response to treatment is not clear cut (https://www.lungcancerjournal.info/article/S0169-5002(12)00583-1/fulltext).
Unfortunately, Rb1 and pRB stains are not used in our pathology department. Significant time would be required to validate these stains with appropriate positive and negative controls. We understand that loss of nuclear expression of Rb proteins can be helpful in the identification of small cell carcinomas or the spectrum of neuroendocrine tumors but we are uncertain as to their relevance in this case.
- What is the opinion of the authors regarding COX2? These points must be discussed by the authors in the manuscript in relation to survival for the results obtained to be applicable.
We have added the following paragraph to the discussion regarding CDX2 (we assume COX2 is a typo):
“CDX2 is a homeobox gene encoding transcriptional factors for intestinal differentiation and in adults is expressed in the nuclei of intestinal epithelial cells. As mentioned previously it is used in pathology as a tissue biomarker for intestinal morphology in patients with CRC and in PEAC. A recent meta-analysis of over 6000 patients with CRC showed that the level of CDX2 expression in stage II and III disease proved to be a strong prognostic factor, leading to 70% lower risk of death [16]. However, to our knowledge no such data exists for patients with PEAC.”
- In figure 2A the authors must show the deviations. Show the values.
These values have been included in panel of the updated figure (now Figure 5A).
- The conclusions must be focused towards a precision diagnosis.
We have updated the conclusions based on the reviewer’s comment:
“Personalized medicine has become indispensable in modern medicine and this is perhaps most evident in the field of medical oncology. Lung cancer is arguably one of the most successful examples of this approach where the discovery of molecular alterations in patients' tumors allowed for the development of targeted therapies. This led to improved survival, better tolerability, and the ability to prognosticate based on the specific molecular pattern of the tumor. Recently a targeted therapy was approved for patients with lung cancer harboring the KRAS G12C mutation, the first successful treatment for KRAS mutated lung cancers. This personalized approach is advancing even further with the ability now to genotype a patient's lung cancer using cell free DNA in patients’ plasma or “liquid biopsies” [27], allowing multiple samples to be taken longitudinally on treatment to detect dynamic changes over time.”
and
“Significantly, the use of exome sequencing contributed an additional personalized dimension to this case and has faciliitated a more accurate and reliable diagnosis.”
Reviewer 2 Report
I think this is an interesting case report. Although there have been several reports of pulmonary enteric adenocarcinoma, but an NGS study of lung adenocarcinoma with that of colon adenocarcinoma is interesting information for oncologist and pulmonary pathologist. However, more information on colon adenocarcinoma is needed. For example: operation year, and the histology on colon adenocarcinoma. If possible, the result of immunostaining (such as CK7, CK20, TTF-1, and CDX2) for colon adenocarcinoma is needed. If the result is positive, it would be good to include a photo with them.Author Response
Reviewer 2
We would like to thank both reviewers for taking the time to review our manuscript and for providing the helpful and encouraging feedback. We have endeavoured to address each comment sufficiently so as to improve the manuscript and we detail our responses below.
I think this is an interesting case report. Although there have been several reports of pulmonary enteric adenocarcinoma, but an NGS study of lung adenocarcinoma with that of colon adenocarcinoma is interesting information for oncologist and pulmonary pathologist.
- However, more information on colon adenocarcinoma is needed. For example: operation year, and the histology on colon adenocarcinoma.
We have included additional information on the colon adenocarcinoma resection specimen including year of resection and histology description to the Case Presentation section:
“She had a past medical history of a pT3N1 colon adenocarcinoma treated with complete resection and adjuvant chemotherapy 13 years previously.”
“We compared this tumor to the patient’s prior colorectal carcinoma which was an invasive moderately differentiated adenocarcinoma with focal mucinous differentiation.”
- If possible, the result of immunostaining (such as CK7, CK20, TTF-1, and CDX2) for colon adenocarcinoma is needed. If the result is positive, it would be good to include a photo with them.
We have included the immunostaining in a new Figure 4 A-D which shows positive CK20 and CDX2 and negative staining for CK7 and TTF-1.
Round 2
Reviewer 1 Report
The authors have correctly answered all the reviewer's questions. The changes are appropriate and the manuscript has improved.